# Real-time structural motif searching in proteins using an inverted index strategy

**Sebastian Bittrich** [1] *, **Stephen K. Burley** [1,2,3,4,5], **Alexander S. Rose** [1]

**1** RCSB Protein Data Bank, San Diego Supercomputer Center, University of California, San Diego, La Jolla, California, USA, **2** RCSB Protein Data Bank, Institute for Quantitative Biomedicine, Rutgers, The State University of New Jersey, Piscataway, New Jersey, USA, **3** Department of Chemistry and Chemical Biology, Rutgers, The State University of New Jersey, Piscataway, New Jersey, USA, **4** Cancer Institute of New Jersey, Rutgers, The State University of New Jersey, New Brunswick, New Jersey, USA, **5** Skaggs School of Pharmacy and Pharmaceutical Sciences, University of California, San Diego, La Jolla, California, USA

* sebastian.bittrich@rcsb.org

**Data Availability Statement:** All relevant data are within the manuscript and its Supporting information files. The source code is available on GitHub (https://github.com/rcsb/strucmotif-search).

## Abstract

Biochemical and biological functions of proteins are the product of both the overall fold of the polypeptide chain, and, typically, structural motifs made up of smaller numbers of amino acids constituting a catalytic center or a binding site that may be remote from one another in amino acid sequence. Detection of such structural motifs can provide valuable insights into the function(s) of previously uncharacterized proteins. Technically, this remains an extremely challenging problem because of the size of the Protein Data Bank (PDB) archive. Existing methods depend on a clustering by sequence similarity and can be computationally slow. We have developed a new approach that uses an inverted index strategy capable of analyzing >170,000 PDB structures with unmatched speed. The efficiency of the inverted index method depends critically on identifying the small number of structures containing the query motif and ignoring most of the structures that are irrelevant. Our approach (implemented at motif.rcsb.org) enables real-time retrieval and superposition of structural motifs, either extracted from a reference structure or uploaded by the user. Herein, we describe the method and present five case studies that exemplify its efficacy and speed for analyzing 3D structures of both proteins and nucleic acids.

## Author summary

The Protein Data Bank (PDB) provides open access to more than 170,000 three-dimensional structures of proteins, nucleic acids, and biological complexes. Similarities between PDB structures give valuable functional and evolutionary insights but such resemblance may not be evident at sequence or global structure level. Throughout the database, there are recurring structural motifs—groups of modest numbers of residues in proximity that, for example, support catalytic activity. Identification of common structural motifs can reveal similarities between proteins and serve as fingerprints for spatial configurations of amino acids, such as the His-Asp-Ser catalytic triad found in serine proteases or the zinc coordination site found in Zinc Finger DNA-binding domains. We present a highly

**Funding:** RCSB PDB is jointly funded by grants to SKB from the National Science Foundation (DBI-1832184, https://www.nsf.gov/), the US Department of Energy (DE-SC0019749, https://www.energy.gov/), and the National Cancer Institute, National Institute of Allergy and Infectious Diseases, and National Institute of General Medical Sciences of the National Institutes of Health (https://www.nih.gov/) under grant R01GM133198. The funders had no role in study design, data collection and analysis, decision to publish, or preparation of the manuscript.

**Competing interests:** The authors have declared that no competing interests exist.

efficient yet flexible strategy that allows users for the first time to search for arbitrary structural motifs across the entire PDB archive in real-time. Our approach scales favorably with the increasing number and complexity of deposited structures, and, also, has the potential to be adapted for other applications in a macromolecular context.

## Introduction

Within proteins, structural motifs are characteristic arrangements of amino residues, which may or may not be near one another in the linear polypeptide chain. They can be described using three properties, including spatial proximity, relative arrangement in three-dimensions (3D), and physicochemical properties [1]. Many structural motifs contribute directly to the biochemical or biological function of a protein, such as the well-known His-Asp-Ser catalytic triad (Fig 1A) found in serine proteases [2]. More complex structural motifs contribute indirectly to function by acting as binding scaffolds (*e.g.*, five residues coordinating two zinc ions within the active center of bovine lens leucine aminopeptidase, PDB ID 1lap [3], Fig 1B). Detection of structural motifs can reveal subtle evolutionary relations between proteins and provide insights into function for as yet uncharacterized biomolecules [1, 4]. In other cases, structural motifs common to two proteins are the result of convergent evolution that allows two polypeptide chains of quite different 3D folds to catalyze very similar, if not identical,

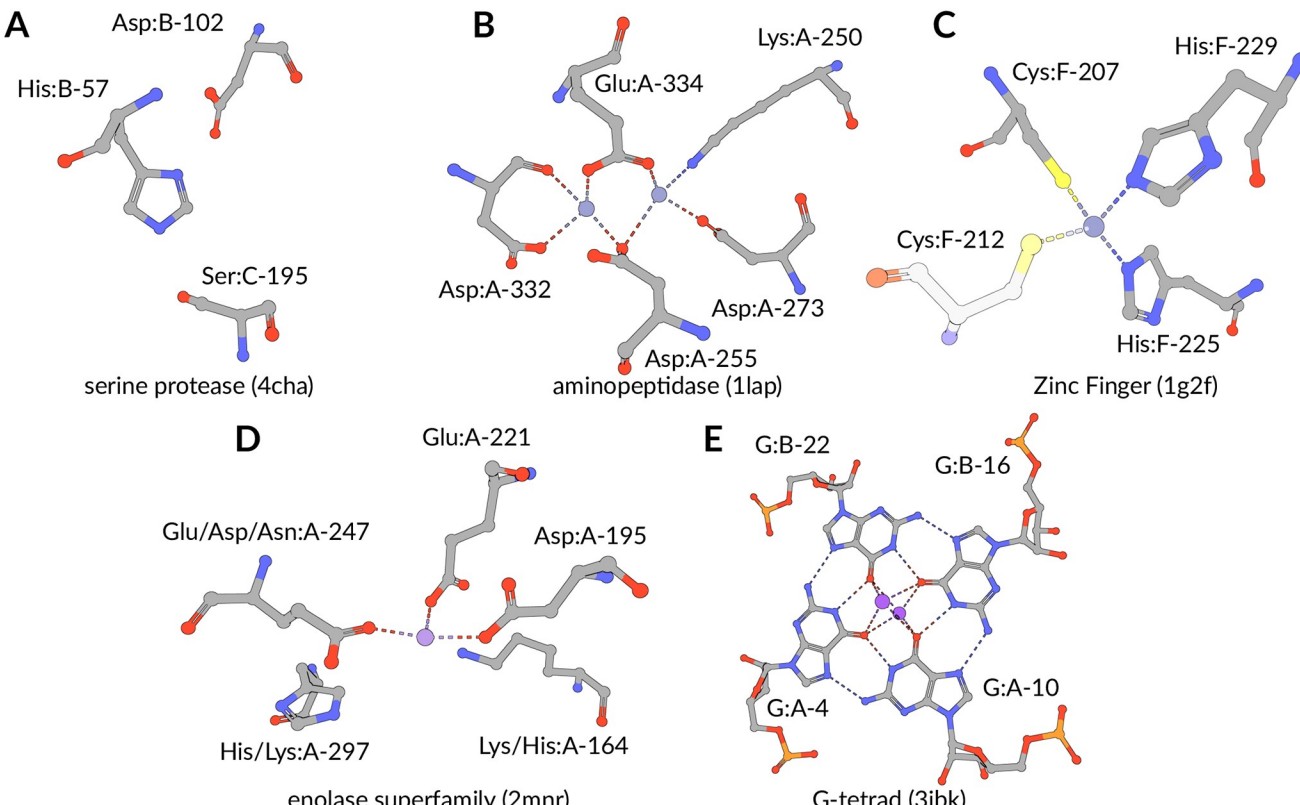

**Fig 1. Structural motif case studies.** (**A**) Active sites of serine proteases can be made up of multiple, distinct polypeptide chains [2]. (**B**) Active site sidechains in leucine aminopeptidases [3] coordinate two adjacent ions. (**C**) Zinc Finger DNA-binding domains [5] are stabilized by zinc ions (N.B.: Cys:F-212 was not used to define the search query.) (**D**) Position-specific exchanges (additional *label_comp_id*) can be used to identify enolase superfamily members accurately [6]. (**E**) RNA G-tetrads can be formed between one, two, or four nucleic acid strands [7]. *label_comp_id*, *auth_asym_id*, and *auth_seq_id* as residue identifiers. Rendering by Mol* [8].

chemical reactions [1]. Thus, identification of structural motifs represents a powerful tool for finding functional similarities between proteins, which may not be evident when relying solely on primary, secondary, tertiary, or even quaternary structures.

Searching for structural motifs can be viewed as a computationally expensive task that requires exhaustive evaluation of many possible combinations in 3D. For a polypeptide chain of length $n$, there are $\binom{n}{k}$ combinations of motifs of size $k$. Conversely, deciding whether or not a particular structural motif is present in a given protein can be formulated as subgraph isomorphism problem, which is NP-complete [9]. It is also challenging to automatically classify structural motifs [1, 4, 10, 11], a task commonly realized by identifying overrepresented structural motifs found within a protein family [10, 11]. An alternative approach involves manual or semi-automatic biocuration by subject-matter experts. For example, the Catalytic Site Atlas gathers such definitions of enzyme structural motifs [12].

Several structural motif search routines have been implemented over the past three decades. They commonly employ geometric hashing techniques or graph theoretical methods (reviews in [1, 4]). Geometric hashing strategies [10, 13–16] describe the relative arrangement between two (or three) residues in a rotation invariant fashion. Typically, geometric descriptors are used to create a reduced hash representation of complex 3D data (*e.g.*, distances between $C_\alpha$ atoms in motif residues). Such simplifications allow geometric hashing approaches to perform computationally expensive tasks only once during a preprocessing step and then reuse the results of this initial step to support rapid query responses [4]. This strategy requires additional storage but permits repeated reuse of the computed reduced low-dimensional representations.

An alternative approach is found in graph theoretic methods [17–19]. Proteins can be represented as graphs with residues being the vertices and edges capturing spatial proximities. A structural motif search can then ask whether or not a corresponding subgraph occurs in a graph defined by the whole protein. In addition, there are relatively efficient combinatorial approaches such as Fit3D [20], which make exhaustive searching feasible by rejecting candidates early in the process. This step is crucial for an exhaustive search, as it would otherwise require evaluation of every structure file in the search space. Uniquely, the Suns [9] tool adapts strategies from web search engines and enables real-time discovery of similar motifs for protein design. However, its speed is achieved by reporting only limited results. This approach is inadequate for more general applications, such as statistical analyses [21, 22].

Given ∼10% year-on-year growth of 3D biostructure data in the PDB [23, 24], practicable search implementations must scale linearly with the size of the search space [25]. For global structure comparison, we recently published an efficient approach using BioZernike moments [25]. Users can perform queries on >170,000 PDB structures and retrieve results instantaneously. There is currently an unmet need for a complementary strategy capable of near instantaneous searching for structural motifs across the entire PDB archive.

Herein, we present a real-time structural motif search algorithm that returns results within seconds using readily available computational resources. A novel indexing strategy, inspired by web search indices, describes the relative spatial arrangement of residue pairs ("words" in a text search context) composed of amino acids and/or nucleotides. The structural motif search problem can then be formulated as search for a collection of residue pairs in a set of protein structures ("documents" in a text search context). Search engines tackle similar problems by creating an inverted index [26] that keeps track of all documents containing certain words. For our motif search problem, such a word-level inverted index keeps track of the polymer sequence positions at which certain residue pairs occur. This strategy enables quick retrieval of residue pairs in a large data corpus akin to the index at the end of a printed volume. It can be used to support real-time structural motif searching across the entire PDB archive and other

large 3D structural data sets. Our approach supports several advanced features, including analysis of motifs distributed across multiple polymer chains, support for nucleic acid motifs and post-translationally modified residues, and position-specific exchanges for analyzing evolutionary relationships. Dissimilarity among ensembles of structural motif hits is quantified using the root-mean-square deviation (R.M.S.D.) measure.

## Results and discussion

### Structural motif searching using inverted indexes

Structural motif searching using an inverted index approach involves creating a lookup table for all arrangements of residues present in the PDB (see Methods). This strategy recasts the search problem to one of loading all occurrences from the inverted index. To achieve this goal, a query motif (Fig 2A) is specified as a collection of residues. The query is fragmented into residue pairs and each pair is represented by a rotation-invariant, symmetric descriptor (Fig 2B) that is extracted from the known coordinates given by the query. The descriptors are based on residue labels (*e.g.*, serine, aspartic acid, and histidine for the catalytic triad), the backbone distance $d_b$ ($C_\alpha$ for amino acids), the sidechain distance $d_s$ ($C_\beta$ for amino acids), and the relative angle $\theta$ defined by the two vectors connecting backbone and sidechain of each residue (see Methods). Exact values are binned for distances (width = 1 Å) and angles (20˚). Similar occurrences of residue pairs are thereby sorted to the same bin.

The inverted index approach supports lookup of all occurrences of a certain residue pair descriptor sharing similar geometric properties (Fig 2C). The result is an exhaustive map of all PDB structures (designated by unique PDB IDs; *e.g.*, 1abc) wherein this residue pair occurs at least once. Individual occurrences are identified by an expression such as A_1–87, wherein A_1 corresponds to *label_asym_id* A with the first assembly generation operation and 87 refers to the residue with *label_seq_id* 87 within a given PDB structure. Thus, our

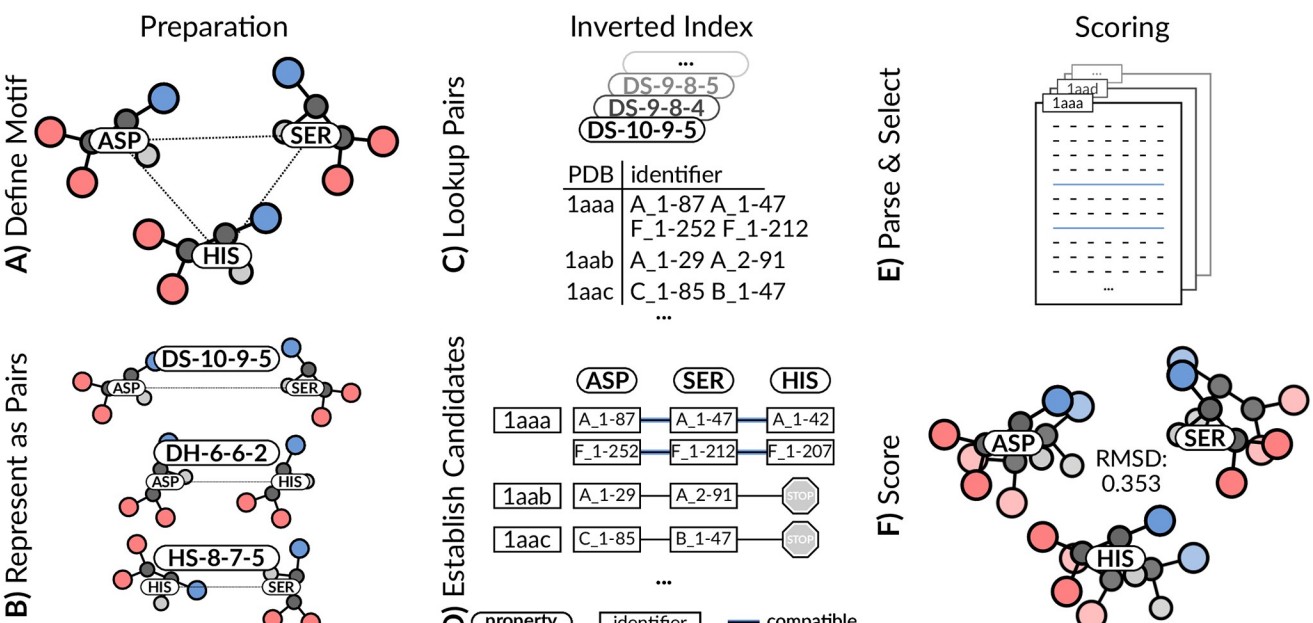

**Fig 2. Structural motif search workflow.** (**A**) Fragmentation into residue pairs. (**B**) Computation of geometric descriptors. (**C**) Inverted index lookup. All similar occurrences are retrieved for each descriptor. (**D**) Checking for correspondence to ensure that candidate resembles query motif. (**E**) Structures not fulfilling requirements are ignored. Only relevant residues are loaded. (**F**) R.M.S.D. measures quantify structural similarity.

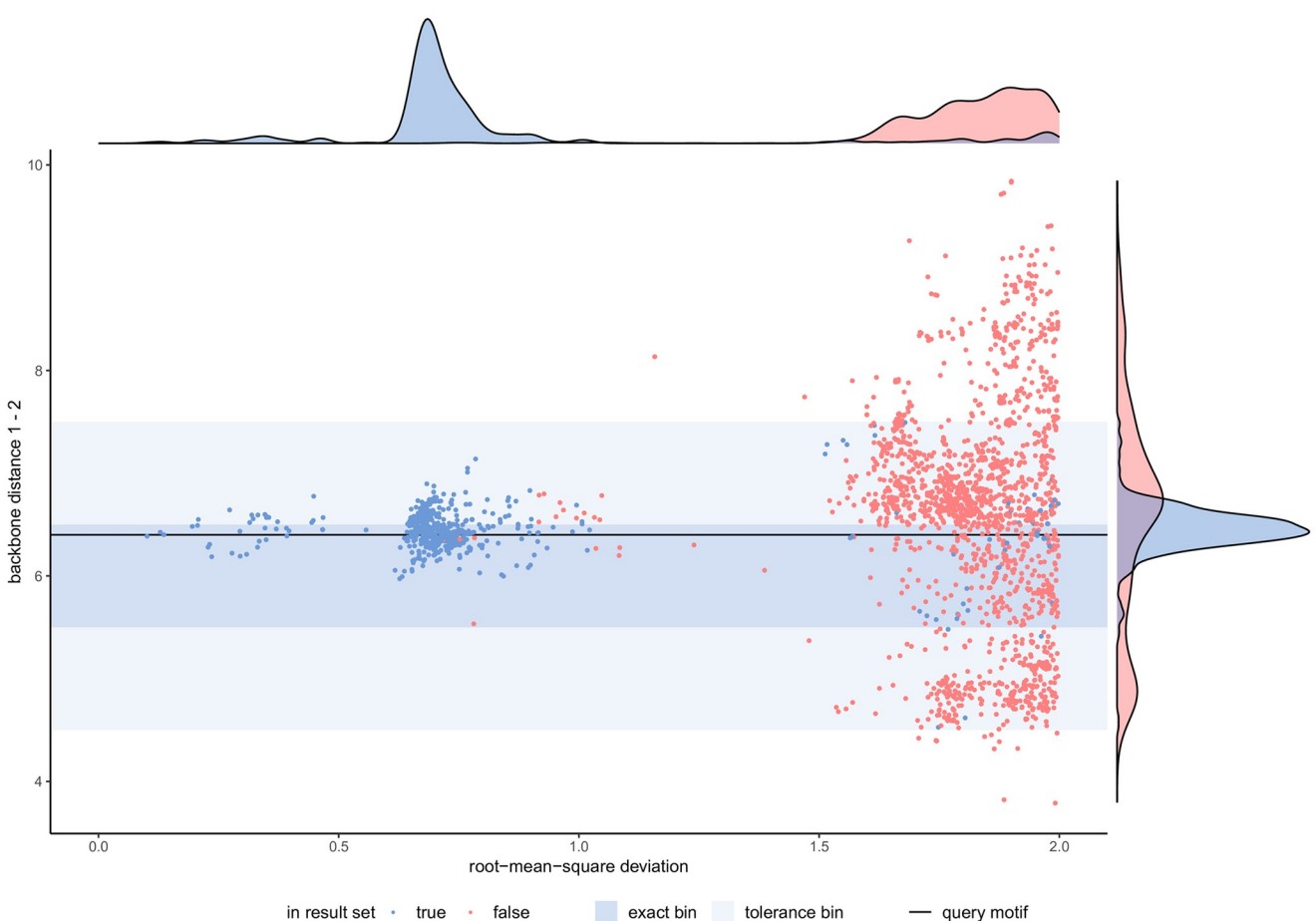

**Fig 3. Sensitivity of geometric descriptors.** Ground truth for catalytic triad query was determined by an exhaustive search routine [27]. Most low R.M.S.D. hits are found by our approach (blue points). Biologically relevant hits exhibit geometric descriptors (area shaded in blue) similar to the query motif (black horizontal line).

approach generates a word-level inverted index (*i.e.*, it reports both the PDB ID and the unique position of its occurrence). The binning strategy for distances (or angles) requires that the lookup surveils neighboring bins (Fig 2C), to ensure that occurrences close to bin boundaries are not lost (Fig 3). By default, a tolerance value of 1 is employed, resulting in lookup in three bins for each geometric descriptor. The inverted index contains information on pairs of residues; thus, partial results have to be combined to represent motifs with 3 or more residues (Fig 2D). Valid candidates are collections of residues that fulfill requirements imposed by the structural motif query. 3D structures in the search space can be ignored if they do not contain all residue pairs present in the query motif. Moreover, all residue pairs of a valid candidate must occur in the correct spatial arrangement. This approach allows us to determine whether or not a certain PDB structure contains a query motif without actually loading any structural data.

For example, no compatible residue pairs occur between histidine and serine (descriptor HS-8-7-5) for PDB IDs 1aab and 1aac during His-Asp-Ser structural motif candidate assembly (Fig 2D). This finding permits rejection of these PDB IDs in all subsequent candidate assembly operations. In practice, qualifying valid candidates allows the structural motif search to be narrowed to hundreds or at most thousands of PDB structures. S1 Table shows how the serine

protease query requires loading of only 4,577 PDB structures, ignoring ∼97% of the PDB archive because the inverted index reported no valid candidates for these structures.

Optionally, structure data is read for candidates that contain residue pairs similar to the query (Fig 2E). Each hit is then aligned to the query motif and its dissimilarity is scored by the R.M.S.D. (Fig 2F). The computed R.M.S.D. can be used for downstream operations such as filtering or sorting of hits. Finally, a result list is composed of all returned hits. See Materials and methods for details.

## Runtime analysis

Table 1 provides results of runtime analyses. Our approach provides results within 1 or 2 seconds and scales linearly with the number of structures in the search space (S1 Fig). The majority of the computation time is devoted to reading the inverted index and qualifying valid candidates. The remainder of the runtime is dedicated to retrieving coordinates and computing R.M.S.D. values for each putative hit.

The interplay between structural motif definition and runtime is complex. Motif size and composition exhibit no clear influence on the runtime. The same is true for the number of hits over which R.M.S.D. values are computed. Most significant in determining runtime is the structure of the corresponding bin of the inverted index (see below). Highly populated bins require more time for input/output and also require more logic operations to qualify valid candidates. In comparison to other methods, our approach is indisputably superior in computational efficiency. It does not resort to reporting only a small number of similar motifs [9], nor does it require that the search space be filtered in advance to eliminate redundancy. Searching the entire PDB is necessary to integrate the structural motif search routine with other RCSB PDB (rcsb.org) search capabilities, such as sequence similarity, text search, or shape search [23, 25, 28]. Moreover, our approach will not collapse under the weight of the relentlessly growing PDB, because its complexity varies linearly with the number of structures in the archive (S1 Fig). Queries with position-specific exchanges allow a set of similar amino acids at the same position (as shown for the enolase superfamily template). Such queries require many more read operations in the inverted index but exhibit only a slightly higher response time (S1 Table). As a comparison to an existing method, S2 Table summarizes the number of results and the required runtime by the Fit3D web server [27].

Individual queries can be processed quickly because the majority of the required computations were performed once during the creation of the inverted index. A full load of 169,117 structures (PDB archive snapshot on 9/25/20) took 3 days and 11 hours. Furthermore, our implementation supports incremental load operations. An incremental load adds the set of

**Table 1. Runtime and sensitivity analyses.**

| Structural Motif | Hits | Time | First FN | R.M.S.D. | FNR <1 Å |
|---|---|---|---|---|---|
| serine protease [2] | 3,498 | 0.92 s | 1hcg | 0.72 Å | 1.86% |
| aminopeptidase [3] | 350 | 0.46 s | N/A | N/A | 0.00% |
| zinc coordination [5] | 1,056 | 0.13 s | 2eou | 0.42 Å | 8.52% |
| enolase superfamily [6] | 288 | 0.36 s | 3mkc | 0.89 Å | 2.33% |
| enolase (exchanges) | 308 | 0.87 s | 3mkc | 0.89 Å | 5.62% |
| RNA G-tetrad [7] | 84 | 1.10 s | 2rsk | 1.53 Å | 0.00% |

Each reported runtime represents the average over 10 benchmark runs. Results are compared to an exhaustive search strategy [27] and the minimum R.M.S.D. of any false negative (FN) hit is reported. The false negative rate (FNR) below 1 Å (*i.e.*, a general cutoff below which we consider hits to be biologically meaningful) is given. S1 Table shows that false negative rate can be reduced at the cost of a moderate runtime increase.

PDB identifiers that were deposited since the last update of the inverted index. One week later (on 10/2/20), the incremental load processed 319 structures in 2 hours and 46 minutes. Following the update, all 169,436 were available in the inverted index.

## Case studies

We exemplify uses of our approach with five previously characterized structural motifs that are well represented in the PDB (Fig 1). These examples support diverse biological/biochemical functions and differ in size and complexity. The catalytic triad found in serine proteases [2] is a frequently showcased structural motif. Some occurrences are difficult to detect because the motif may be distributed among multiple polypeptide chains. The active site of leucine aminopeptidase [3] is a larger motif encompassing five residues, collectively responsible for coordinating two zinc ions. The $His_2/Cys_2$ Zinc Finger motif [5] provides an alternative example of a structural motif responsible for metal binding that stabilizes the structures of many DNA-binding domains found in many eukaryotic transcription factors. Complex evolutionary aspects can be represented by structural motifs with position-specific exchanges [20] as seen for the enolase superfamily [6]. G-tetrads are a prominent nucleic acid association motif [7].

We assessed the false negative rate by comparison to an established, exhaustive search strategy represented by Fit3D [20], a method based on rigid alignments that scores hits by R.M.S.D. values. We filtered the Fit3D result list for R.M.S.D. values <1 Å to identify true positive hits that our method should report in any case (these hits may not be biologically functional, but should be regarded as promising candidates meriting closer inspection). Our method also finds additional hits because the Fit3D web server operates on a redundancy filtered version of the PDB archive.

**Serine proteases: Detection of the catalytic triad.** Many hydrolases use a serine nucleophile during catalysis. Canonical serine protease catalytic triads are composed of His, Asp, and Ser residues (Fig 1A—PDB ID 4cha—His:B-57, Asp:B-102, Ser:C-195). They typically occur within two polypeptide chains, because many proteases are initially made as zymogens that require activation by proteolytic processing [2] to prevent uncontrolled digestion of proteins within the cell. A His, Asp, and Ser catalytic triad query based on the configuration of these three residues in PDB ID 4cha returns 3,498 hits in ~0.9 s. When compared to Fit3D [20], our inverted index method gave a false negative rate of <2%. Our approach is independent of the number of polymer chains over which the catalytic triad structural motif is distributed. We successfully detected both known examples involving three polypeptide chains, including trypsin (PDB ID 1ept—His:B-41, Asp:C-52, Ser:A-40—R.M.S.D. = 0.3 Å) and thrombin (PDB ID 2hnt—His:C-26, Asp:D-51, Ser:B-43—R.M.S.D. = 0.3 Å). Both proteins support proteolytic activity. In these rare cases, zymogen activation by peptide cleavage yields a structural catalytic motif distributed over three chains instead of the canonical two chains. This example shows that our approach comes very close to replicating the results of an exhaustive search strategy in less than a second, even for a structural motif that is highly abundant in the PDB archive. For comparison the runtime of the exhaustive search [27] for the catalytic triad benchmark on server hardware required 131 s. We used the least restrictive *BLASTe-80* target list (36,213 structures evaluated), Fit3D returned 538 matches below 1 Å R.M.S.D. Our approach returned 2,976 hits with R.M.S.D. <1 Å after evaluating the entire PDB archive. It also successfully identifies inter-chain arrangements of residues matching the query, and the method is agnostic as to polymer type(s) (protein *versus* nucleic acid) and the presence of chemical modifications (*e.g.*, phosphorylation).

Fig 3 depicts how R.M.S.D. values relate to the significance of a given hit detected by our inverted index method. The density distribution shows two distinct sets of hits: one with <1.0

 

Å R.M.S.D. and the other with >1.5 Å. Hits with high R.M.S.D. values likely encompass arrangements of histidine, aspartic acid, and serine that happen to be in proximity but are unable to support serine protease enzyme activity. The ensemble of hits with higher R.M.S.D. values also show higher variance with respect to the discussed geometric descriptors. In contrast, hits with lower R.M.S.D. values all exhibit geometric properties that resemble the query motif (horizontal black line). A further speedup of our method can be achieved by identifying candidates that yield low R.M.S.D. values and ignoring the remainder. The tolerance parameter specifies how much deviation from the query motif is tolerated (area shaded in blue). We used R.M.S.D. <1 Å as a preliminary cutoff for the remaining cases discussed in this paper.

Additionally, we investigated how well our inverted index method coincides with functional annotation resources (S3 Table). Therefore, we collected PDB structures that share an Enzyme Commission number (EC 3.4.21.1), an entry in the Catalytic Site Atlas (M-CSA ID 387, [12]), or a PROSITE pattern (PS50240, [29]) with PDB ID 4cha from which the query motif was extracted. For all resources, >90% of hits are found with default parameters. Higher tolerance values result in complete coverage of EC 3.4.21.1 but do not result in substantially higher coverage of M-CSA ID 387 or PS50240. The functional annotations considered are based on homology or sequence patterns and include some occurrences that may not be functionally relevant. For example, the structure of PDB ID 1a7s aligns well (R.M.S.D. = 0.717 Å) but the active site in question exhibits 2 amino acid substitutions. Analogously, the active site of PDB ID 1bio contains a covalently bound inhibitor that may cause an atypical conformation of His:A-57 [30]. Sequence-based methods are orthogonal to structure-based ones, thus, it is advantageous to use multiple resources to screen for protein function [19].

**Aminopeptidase: Retrieval of all occurrences of a rigid motif.** Aminopeptidases play important roles in protein degradation by removing residues from the N-terminus of polypeptide chains [3]. Bovine leucine aminopeptidase (BLLAP) is a homohexameric enzyme with $3_2$ quaternary symmetry. The active site of BLLAP contains two adjacent zinc ions separated by ~2.9 Å and coordinated by the sidechains of five conserved residues Lys, Asp, Asp, Asp, and Glu (Fig 1B—PDB ID 1lap—Lys:A-250, Asp:A-255, Asp:A-273, Asp:A-332, Glu:A-334). This five-residue query could be executed in <0.5 s, approximately half the time required for the catalytic triad search. Query motifs with more than three residues are simplified by a minimum spanning tree approach (see Methods), which removes cycles from the graph defined by the query. This approach results in $n - 1$ constraints with $n$ referring to the number of residues. For BLLAP, all 9 structures curated with UniProt ID P00727 in the PDB were detected with this structural motif search. The remaining 48 hits (R.M.S.D. <1 Å) represent leucine aminopeptidases from other organisms or leucine-aminopeptidase-like enzymes, such as the bottromycin maturation enzyme (PDB ID 5lhj, R.M.S.D. = 0.4 Å, sequence identity to BLLAP = 35.3%). One PDB structure similar in sequence to BLLAP was not detected by our query search due to a deviating arrangement of motif residues (motif R.M.S.D. = 2.2 Å). PDB ID 3pei a cytosolic aminopeptidase from *F. tularensis* (the causative agent of tularemia) is similar to BLLAP in both sequence (amino acid identity = 38%) and 3D structure (R.M.S.D. = 1.0 Å for 291 alpha carbon atomic pairs). It is also a homohexamer with the same $3_2$ quaternary symmetry as BLLAP. This particular structure shares the same five active site residues used for the motif search query, albeit with slight differences in their relative arrangement. (N.B.: Fit3D also failed to detect PDB ID 3pei.) The *F. tularensis* aminopeptidase structure was determined in the absence of divalent metal ions, which may explain the observed structural differences between the two active sites *versus* BLLAP. Alternatively, the prokaryotic enzyme may be dependent on an alternative choice of divalent metals. The foregoing discussion serves to underscore the fact that motif definitions vary in their utility and it may be challenging to find optimal representations for more flexible motifs [31].

**Zinc Fingers: Query definition is key.**   Eukaryotic transcription factors often contain His$_2$/Cys$_2$ Zinc Finger domains (Fig 1C—PDB ID 1g2f—Cys:F-207, Cys:F-212, His:F-225, His:F-229). These motifs are composed of two cysteine and two histidine residues, which stabilize a small $\beta\beta\alpha$ domain structure that envelopes and coordinates a single zinc ion [5]. In the absence of the zinc ion, these domains do not adopt compact, folded structures and are incapable of binding DNA. Our experience with the His$_2$/Cys$_2$ motif reflects the importance of the way in which the query is constructed. We found that the cysteine corresponding to F-212 in the query search occurs commonly in a distinct orientation *versus* the original motif definition. Consequently, the number of false negatives identified in this four-residue motif definition was higher than expected ($\sim$67.0%). To overcome this subtle structural variation within the His$_2$/Cys$_2$ Zinc Finger family of DNA-binding domains, we employed a simplified query definition omitting the cysteine corresponding to F-212. S2 Fig shows the variability of the angle descriptor between residues 1 and 2 that gave rise to the unexpectedly large number of false negatives. The first false negative (PDB ID 2elv) is depicted in S3 Fig for illustrative purposes. The polypeptide chain backbone geometry in the vicinity of Cys:F-212 also varies within this DNA-binding domain family.

A simplified 3-residue search query (PDB ID 1g2f—Cys:F-207, His:F-225, His:F-229) yielded more hits and compares much more favorably to the results of an exhaustive search (false negative rate = 8.5%). The runtime for the three-residue query increases only slightly *versus* the four-residue query, which can be attributed to more structures being included in the R. M.S.D. value computation. These results show that too rigorous a query definition can lead to an unacceptable number of false negatives. In contrast to the histidine residues that are part of the $\alpha$-helix, the cysteine residues corresponding to Cys:F-212 occur exclusively in random coil loop segments [5], which may exhibit greater structural flexibility than residues present in defined secondary structural elements. This challenge could be mitigated by defining suitable curated queries for popular motifs within the production version of this method to be implemented on the RCSB Protein Data Bank website (rcsb.org) or by providing the information as a dedicated public resource akin to the Catalytic Site Atlas [12].

The efficiency of our method allows us to recommend that users vary search motif definitions and seek consensus among multiple runs of similar but subtly different queries. To underscore this point, we investigated whether the simplified 3-residue search query can be refined further. The low runtimes of our method allow optimization of query definitions by using all accepted hits of an initial run as query definitions for individual, subsequent runs. Some query results will return fewer hits than the initial query, while others may report more or possibly different hits. For the zinc finger motif, more than 1,000 queries were processed within 111 s. The query based on PDB ID 2emb (Cys:A-15, His:A-31, His:A-35) returned the most hits and more than doubled the size of the result set to 2,261. PDB ID 5yef (Cys:D-36, His:D-49, His:D-54) features the largest addition of 1,571 previously unidentified hits, but also misses 676 hits that were captured by the initial query motif. PDB ID 5c8t (Cys:D-280, His:D-258, His:D-265) returns the smallest result set with only 208 hits. All of these motifs feature a coordinated zinc ion in the PDB structure. This experience demonstrates the importance of the query definition. In cases where exact geometry is subordinate, it may be beneficial to search for multiple query definitions and merge these results to produce a comprehensive, non-redundant set of PDB structures containing the structural motif in question.

**Enolase superfamily: Efficient searching with position-specific exchanges.**   The enolase superfamily is a group of proteins diverse in sequence, yet largely similar in 3D structure that all catalyze abstraction of a proton from a carboxylic acid [32]. The structural motif supporting this catalytic function [6] is represented in PDB ID 2mnr (Fig 1D—Lys/His:A-164, Asp:A-195, Glu:A-221, Glu/Asp/Asn:A-247, His/Lys:A-297). This particular case is one of the more

challenging for motif searching in 3D. Isofunctional exchanges between histidine and lysine have been observed for the first and last position. Similarly, the glutamic acid at A-247 can be substituted by aspartic acid or asparagine [6]. When no exchanges are considered, the simplest possible query returned 288 hits within <0.4 s. When position-specific exchanges are incorporated within the query, the number of hits with R.M.S.D. <1 Å increases to 308 (computation time <0.9 s). Additional hits with exchanges exhibit R.M.S.D. values ~1 Å. As described earlier, increased tolerance values result in retrieval of all hits (S1 Table). An increased tolerance value results in no false negatives with R.M.S.D. <1 Å for the query with position-specific exchanges. The enolase superfamily example demonstrates that our method can process even combinatorially complex queries very efficiently. The number of candidates to evaluate increases dramatically when increased tolerance values and position-specific exchanges are involved at only modest cost in terms of computational time (Table 1).

**RNA G-tetrad: Nucleotide motifs in biological assemblies.** G-tetrads are a common nucleic acid association motif. They are composed of guanine and stabilized by Hoogsteen base pairings (Fig 1E). The four O6 oxygen atoms coordinate monovalent ions, such as $K^+$, and individual tetrads tend to be stacked one atop the other [7]. A query for the G-tetrad motif takes ~1 s to complete and returns 84 hits. Interestingly, some G-tetrads arrangements were detected within larger assemblies. For example, PDB ID 3mij (A-4 and A-10) with a R.M.S.D of ~0.7 Å was detected in a telomeric RNA G-quadraplex. These results document that our approach provides support for facile searching of structural motifs in nucleic acids, and also indexes occurrences in homo- and heteromeric biological assemblies, including protein-nucleic acid complexes.

**Structure of the inverted index.** We constructed the inverted index for the PDB as of 2/17/20 encompassing 160,467 distinct structures. The size of the index of amino acid pairs is ~55 GB, distributed among 239,034 bins (unique combinations of amino acids, distances, and angles). The index contains 6,814,159,549 residue pairs with distance $d_b$ of up to 20 Å, with the largest 21,487 bins representing >50% of all occurrences. Each bin contains an average of 28,514 occurrences (positions referenced wherein this residue pair is observed) distributed over an average of 11,965 PDB structures. The ten largest bins (Table 2) contain combinations of alanine, glutamic acid, glycine, and leucine residues. The most frequent residue pairs tend to capture sequence neighbors (as indicated by alpha carbon separations of ~4 Å). Other common residue pairs are those with distances $d_b$ near the cutoff value of 20 Å. In general, bins are of particular interest when they contain an elevated number of entries reflecting function (*e.g.*,

**Table 2. Ten most abundant residue pairs in the inverted index.**

| Residue Pair Descriptor | # Structures | # Occurrences |
|---|---:|---:|
| AL-4-5-4 | 110,080 | 903,128 |
| GL-19-19-2 | 103,954 | 555,431 |
| AL-10-11-5 | 103,570 | 643,736 |
| EL-4-5-4 | 103,470 | 623,808 |
| EL-5-7-6 | 103,351 | 582,756 |
| AE-4-5-4 | 103,191 | 639,248 |
| EL-10-11-5 | 102,304 | 544,341 |
| GL-18-18-2 | 101,624 | 510,194 |
| GL-19-19-3 | 101,494 | 520,548 |
| AL-19-19-2 | 101,308 | 612,042 |

Sorted by number of structures containing this residue pairs. Only amino acid pairs were considered.

catalytic triads in serine proteases [2]) or non-covalent interactions responsible for protein structure stabilization.

As described above, our inverted index approach dramatically reduces the search space even for individual residue pairs. Even the largest bin AL-4-5-4 (Table 2) eliminates approximately one third of the PDB archive from consideration during a search involving that particular combination of residues. Inverted index-based queries for functional motifs usually consist of multiple residue pairs. In these cases, the inverted indexing strategy pinpoints a few hundred to thousands of PDB structures that contain the query (S1 Table). The computational economy of this strategy underpins the speed of our approach, which is particularly relevant given the increasing complexity of more recent PDB depositions [23]. We assessed the impact of the increasing size of the PDB by comparing the current PDB (∼170,000 structures) to the holdings of the archive at the end of 2012 (roughly half the size, 78,237 structures). The 2012 inverted index contains 2,483,893,161 residue pairs (∼37% of the 6,814,159,549 residue pairs for the current PDB). Between 2012 and 2020, the number of PDB structures increased by ∼105%, while the size of the inverted index increased by ∼174%. The fact that the inverted index did not grow in strict 1:1 proportion with the growth in the number of new structures reflects the fact that the average size of a structure deposited to the PDB has been increasing year-on-year [33]. The average PDB structure size (number of residues) rose from 620 to 885 amino acids, comparing the 2012 archive snapshot to the entire contents of the current PDB.

## Conclusion

The search for structural motifs is a computationally challenging task because it involves evaluating large numbers of possible combinations. We have developed a robust, efficient method that utilizes a reduced representation of pairs of residues with two distance descriptors and one angle descriptor. Our approach enables composition of an inverted index that groups similar occurrences together and is independent of the number of residues comprising each structure, which is essential with increasing complexity of recently deposited structures [23, 28]. The inverted index is used to retrieve the position of all occurrences similar to a query within seconds. R.M.S.D. values can be computed for all occurrences. We provide a feature-rich implementation (supporting multiple chains, bioassemblies, and position-specific exchanges), which is available as an open-source project (github.com/rcsb/strucmotif-search) and will be used to augment the search capabilities provided by RCSB PDB [23, 28] on rcsb.org.

We set out to augment the RCSB PDB search capabilities by developing a scalable structural motif search strategy that goes well beyond existing technologies. Our implementation yielded an efficient and accurate processes that supports even quite sophisticated structural motif searches across the >170,000 structures currently represented in the PDB archive. Run times will scale linearly with the growth of the archive. The new tool will enable rapid testing of hypotheses (*e.g.*, by defining varying motif definitions [31]) for the millions of users who frequent the RCSB PDB website (rcsb.org) on an annual basis. This work builds upon previous experience and ongoing developments at RCSB PDB, all of which all aim at ensuring efficient management of the ongoing 3D data deluge [8, 25, 34, 35].

## Materials and methods

### Amino acid and nucleotide representation

During a structural motif query, the labels of the residues are known. In case of the catalytic triad, the query encompasses one histidine, one aspartic acid, and one serine. Thus, the problem can be simplified by ignoring 17 of 20 amino acids. Furthermore, residues occur in a specific distance from one another and their arrangement is constrained by the requirement to

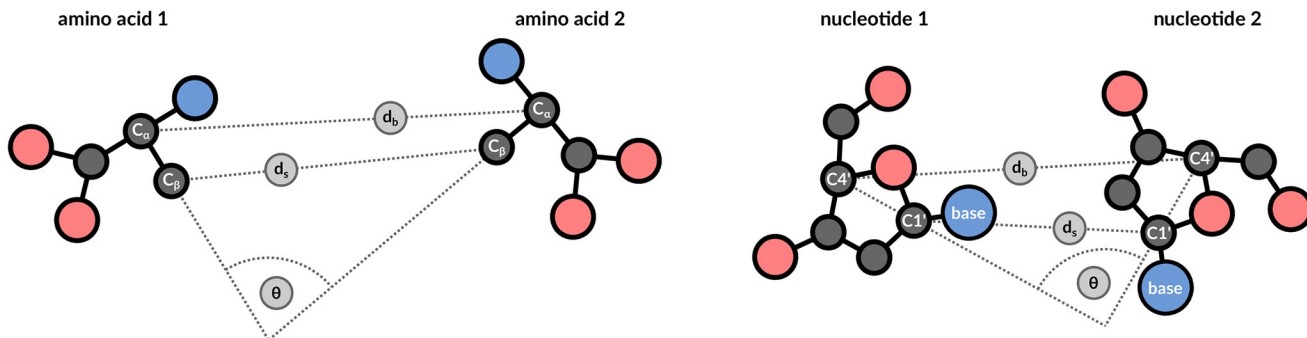

**Fig 4. Representation of residue pairs.** Residue pairs are represented by 3 descriptors that are transformation invariant: backbone distance $d_b$, sidechain distance $d_s$, and angle $\theta$. This constitutes a compact representation of residue pairs and enables quick retrieval of similar pairs.

adapt a functional conformation. We determine three additional properties of each residue pair in the query. We represent amino acids and nucleotides in a generic manner (Fig 4), wherein all amino acids are typified in a comparable way. We assume that the most valuable information is provided by the position of functional groups in the amino acid sidechain. Therefore, we try to balance the influence of backbone and sidechain atoms to accommodate cases with position-specific exchanges. For example, it is difficult to represent the sidechains of a tryptophan and an alanine in a comparable way. Consequently, we choose to represent residues by backbone and sidechain coordinates. For amino acids, we consider $C_\alpha$ as backbone and $C_\beta$ as sidechain representative, respectively. In case of glycine, the $C_\beta$ atom is approximated by superimposing coordinates of a prototypic L-alanine. For nucleotides, we utilize $C4'$ as backbone and $C1'$ as sidechain representative. We represent query motifs and database structures by identifying all residue pairs that are less than 20 Å apart with respect to their backbone distance $d_b$. Motifs can have an arbitrarily large extent as long as they are composed of residue pairs with less than 20 Å distance. We chose a threshold of 20 Å as this can comfortably represent all structural motifs commonly found in PDB structures.

## Geometric description

Backbone distance $d_b$, sidechain distance $d_s$, and the angle $\theta$ can be used to describe the geometric arrangement of residue pairs in the search space (Fig 4). Target structures contain a hit when they contain residue pairs with the same properties as the query motif. We employ a binning approach to assess similarity at residue pair level. Distances are binned into 1 Å groups and angles are grouped by 20° intervals. Our binning approach allows to represent the characteristics of each reside pair by a single integer value that captures both residue types, the $d_b$ bin, the $d_s$ bin, as well as the $\theta$ bin. The presented geometric descriptors are symmetric, as in, independent from the sequence in which both residues appear. For increased storage efficiency, residue type information of descriptors is sorted lexicographically (any residue pair of an alanine and a cysteine is represented by 'AC', there is no bin 'CA'). Similar residue pairs are sorted into the same bin during index creation. Searching requires reading all occurrences registered in a particular bin to identify all residue pairs relevant for a given query. Every lookup operation should encompass reading of neighboring bins, otherwise highly similar residue pairs may result in false negatives because they are separated by a bin border. We chose bin sizes to capture the majority of an exhaustive search strategy as represented by Fit3D [27]. We provide the option to increase the tolerance parameter, which will report more hits at the cost of longer computation time.

## Storage of pair occurrences

The inverted index is implemented by a file system-based approach. Each bin is represented by a dedicated file, which maps between PDB identifiers as keys and an array of residue pair positions as values. Occurrences are identified by *label_asym_id*, assembly generation operator, and *label_seq_id* of the residue in the protein of origin. In other words: This constitutes a word-level inverted index, whereby words are pairs of residues in a certain arrangement and documents are PDB entries. The inverted index readily provides the sequence position of all residues, making retrieval of individual residues convenient and fast. All information is stored in a custom binary data format using the MessagePack codec.

## Assembly of residue pairs into candidates

Speed and sensitivity are prime reasons to minimize the number of operations on the inverted index. Every lookup operation requires I/O time. In addition, more constraints are introduced when all residue pairs of the query are required to be present in a protein structure. This may lead to an elevated number of false negatives. The constraints implied by the residue pairs of the query are transitive (given a constraint between residues A and B as well as B and C, then there is also some constraint on A and C). Therefore, it is advantageous to compose a list of query residue pairs that is as sparse as possible while avoiding specious structures. Motifs with 4 or more residues are pruned by determining the minimum spanning tree of residue pairs using Kruskal's algorithm. Only the selected residue pairs are used to lookup occurrences and perform the search. Candidates that fulfill the query are determined by reading all occurrences for each residue pair. Within the given tolerance range, all residue pair occurrences are pooled together into a map (PDB identifiers as key, collections of occurrences in each structure as value). Valid candidates are identified by enforcing that a certain PDB entry contains all residue pairs specified by the query. Furthermore, the individual residue pairs have to be connected, meaning that they are located close together and not scattered throughout the structure. This requirement is met by testing whether the graph defined by the query motif is isomorphic to any graph present in the PDB entry of interest.

## Management of structure data

The inverted indexing strategy and subsequent screening for candidates reduces the number of coordinate files to assess to hundreds to thousands. During code testing, we found that reading structure data even on this scale accounts for the majority of computation time of each run. We omitted hydrogen atoms and sidechains with multiple conformations, where present (first locations were retained in these cases). Non-polymer groups such as water, ions, and ligands were removed. The precision of the atomic coordinates in Å units was decreased to 1 decimal place (*versus* the normal 3), which has a minimal effect on the computed R.M.S.D. values and eases storage requirements (and therefore time spent on I/O operations). We found that even efficient serialization strategies for compressing macromolecular data such as BinaryCIF [36] or MMTF [34, 35] can create performance bottlenecks. In our prototype implementation, we addressed this challenge by using a database that allows the direct retrieval of individual residues (without reading information on other residues). This approach is particularly beneficial for large ribosome structures or viral complexes. Another possible solution would be to compute alignments (with corresponding R.M.S.D. value and transformation matrix) only when explicitly requested by the user.

## Scoring and significance of hits

The inverted index simplifies the R.M.S.D. calculation by providing the correspondence between residues of query and hit. Otherwise, all combinations of ambiguous labels would need to be evaluated (as is the case for the three aspartic acids in the aminopeptidase example). When position-specific exchanges occur at a position, we form the intersection of atom names and compute the R.M.S.D. for that subset of compatible atoms. No efforts were made to find correspondence between ambiguously labelled atoms [37] (*e.g.*, $C_\delta$ and $C_\epsilon$ atoms of tyrosine). Additionally, it is possible to specify which atoms will be used to compute R.M.S.D. values. An all-atom alignment can put too much emphasis on backbone atoms when chain directions differ, or a certain functionality is exclusively realized by sidechain atoms. In such cases, it may be beneficial to align only sidechain atoms or the atoms used to define the geometric descriptors. The computational load to compute the R.M.S.D. is negligible when a quaternion-based solution [38] is applied. The R.M.S.D. values enable sorting of hits by dissimilarity.

It would be desirable to quantify the significance of hits: *i.e.*, assess the probability that the corresponding hit is the result of a random arrangement of amino acids and not biologically meaningful. Statistical models try to address this problem [21, 22], however no widely accepted solution exists and, maybe, it is even impossible to compute the significance of structural motifs in a truly objective way [1]. We assume that the best way to assess significance is still expert knowledge. The newly added visualization capabilities of Mol* [8] and the cross-references provided by the RCSB PDB web page are an excellent starting point to put a notable hit into further context.

## Visualization of hits

The challenges in significance assessment emphasize the importance of 3D visualization combined with expert knowledge. Visual inspection of whether the protein fold of a structure motif search hit corresponds to that of the query protein. Other characteristics to investigate include solvent exposure of a hit essential for most enzyme active sites and the presence of ligands or ions. The prototype implementation (motif.rcsb.org) is based on the NGL viewer [39]. The prototype allows the definition and submission of custom queries. A list of all results is presented, and the user can align individual motifs to the query as well as align the complete structure that contains query or motif.

## Benchmarking setup

Runtime measurements were performed on a 3.2 GHz Intel Core i7 CPU with 12 cores, 16 GB memory, and macOS. The inverted index and the coordinate database are stored on an SSD. Benchmarks of Java implementations were executed using JMH Java Benchmark Harness using Oracle JDK (HotSpot) 11.0.4. There were some difficulties pertaining benchmarking the search routine. Some time is required to warm-up and optimize the code just-in-time. Therefore, 5 warm-up and 10 measurement iterations performed.

## Exhaustive search for structural motifs

We assessed whether our approach finds the same set of relevant hits (*i.e.*, with a R.M.S.D. value <1 Å) as an established, exhaustive search strategy. The Fit3D web server [27] was used to retrieve a result list for each query motif (Fig 1) using the *BLASTe-80* target list provided by Fit3D. The R.M.S.D. of the first false negative hit was determined with the overall false negative rate (given as the number of hits reported by Fit3D but missing in result set of our method, divided by the total number of Fit3D hits with R.M.S.D. <1 Å).

We do not discuss false positives (*i.e.*, hits found by our method but not by Fit3D) as they are merely the result of the mandatory redundancy filtering by Fit3D or recent additions to the PDB archive. Furthermore, Fit3D allows hits in regions composed solely of alpha carbon atoms. Such hits are likely false positives and cannot be found with our approach because no beta carbons exist to compute descriptors. Therefore, we removed them from the exhaustive list. This methodology was also used to prepare Fig 3. For the G-tetrad motif, we submitted a custom target list of all RNA sequences in the PDB archive.

## Supporting information

**S1 Fig. Impact of archive size on runtime.** Archive was sampled. Runtime for the catalytic triad query increases linearly with archive size.
(TIF)

**S2 Fig. Sensitivity of geometric descriptors for Zinc Finger motif.** A common motif definition, that encompasses four residues, led to an unacceptable number of false negatives. Cysteine F-212 is structurally variable and most hits were not detected when the angle $\theta$ between residues 1 and 2 was below 70˚.
(TIF)

**S3 Fig. Rendering of the first false negative for Zinc Finger motif.** The structurally flexible cysteine at A-15 in PDB ID 2elv (colored in green, motif in light grey) causes this hit to be a false negative. The vector between alpha and beta carbon is orthogonal to that of the query motif.
(TIF)

**S1 Table. Tolerance analysis.** Higher tolerance values lead to higher runtimes. Furthermore, increased tolerance values reduce false negative rate (in comparison to an exhaustive search strategy [20]). We consider hits below 1 Å as biologically meaningful. A tolerance value of 3 does not miss any hits below the threshold.
(XLSX)

**S2 Table. Runtime analysis for Fit3D.** Number of returned hits and runtimes as provided by the Fit3D web server [27]. We consider hits below 1 Å as biologically meaningful. Fit3D returns fewer hits due to redundancy filtering (*BLASTe-80* target list).
(XLSX)

**S3 Table. Sensitivity analysis by functional annotation resources.** EC numbers, Catalytic Site Atlas [12], and PROSITE [29] identified PDB structures with the same functional annotation as the reference structure of the serine protease example (PDB ID 4cha).
(XLSX)

## Acknowledgments

First, we thank the more than 40,000 structural biologists who have deposited structures to the PDB since 2000. We also thank Florian Kaiser for sharing his expertise on structural motif searching and for providing the abstract depiction of amino acids and nucleotides, Michael Schroeder for noting the connection between motif definition and motif search, and gratefully acknowledge contributions from all members of the Research Collaboratory for Structural Bioinformatics PDB past and present and our Worldwide Protein Data Bank partners.

## Author Contributions

**Conceptualization:** Sebastian Bittrich, Alexander S. Rose.

**Funding acquisition:** Stephen K. Burley.

**Methodology:** Sebastian Bittrich, Alexander S. Rose.

**Software:** Sebastian Bittrich.

**Supervision:** Stephen K. Burley, Alexander S. Rose.

**Visualization:** Sebastian Bittrich, Alexander S. Rose.

**Writing – original draft:** Sebastian Bittrich.

**Writing – review & editing:** Sebastian Bittrich, Stephen K. Burley, Alexander S. Rose.

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
