## [Decision Letter · Decision Letter 0]

5 Oct 2020

Dear Bittrich,

Thank you very much for submitting your manuscript "Real-time structural motif searching in proteins using an inverted index strategy" for consideration at PLOS Computational Biology.

As with all papers reviewed by the journal, your manuscript was reviewed by members of the editorial board and by several independent reviewers. In light of the reviews (below this email), we would like to invite the resubmission of a significantly-revised version that takes into account the reviewers' comments.

While I ask you to respond carefully to all reviewers' comments, I would like you to pay particular attention to those of Reviewer #2 and in particular to the request for a more extensive validation of the method.

We cannot make any decision about publication until we have seen the revised manuscript and your response to the reviewers' comments. Your revised manuscript is also likely to be sent to reviewers for further evaluation.

Sincerely,

Marco Punta

Associate Editor

PLOS Computational Biology

Arne Elofsson

Deputy Editor

PLOS Computational Biology

Reviewer's Responses to Questions

**Comments to the Authors:**

Reviewer #1: The paper describes and implements a novel search strategy to query structural motifs across the entire PDB in <2s. The authors describe an ‘inverted index approach’, where the speed efficiency comes from only loading/aligning PDB structures containing residue pairs found in the structural motif. This is achieved by generating a lookup table which details all amino acid pairs (characterised by amino acid types, backbone distance, side chain distance and relative angle – with bins of 1A/20 degrees) found in the PDB, and the corresponding PDB structures they are present in. Pairs of amino acids contained in the query structural motif can then be looked up, and PDB ID’s containing these amino acid pairs identified. These PDB files can then be aligned with the query motif for RMSD calculations. Five case studies are provided, with motifs of 3-5 amino acids in length, and one nucleotide motif. This tool would be useful for the identification of common motifs to suggest protein functionality, or to detect protein similarities.

The strategy is faster than existing structural motif searches, with no clear impact of motif size or composition on run time - which is mostly dependent on the number of PDB entries in the query bin. Comparisons against the Fit3D web server are made. The approach scales linearly with PDB size; important given the increasing number of structures in the PDB. The server is easy to use and allows easy selection of any residues for a motif. The case study of serine protease shows motif identification across multiple polypeptide chains, and the enolase case study includes searching for position-specific amino acid changes. The rate of false negatives is generally low, particularly when tolerance is increased (which does increase runtime), other than when a query residue is in a different orientation to the typical motif - a four-residue zinc finger motif gave a 67% false negative rate. The inverted index size of 39.4GB seems manageable.

Major comments

- Include the time taken to generate the inverted index, it will presumably need regular updating.

- Github page with software is not currently available

Minor Comments

- More detail on the how the false negative rate was calculated and how common it is to get a high false negative rate as seen with zinc finger motif

- Include the time taken to generate the inverted index, it will presumably need regular updating.

- Grammatical/punctuation etc errors:

o Management of structure data paragraph: “We this issue by a database”

o Structure of the Inverted Index paragraph: “multiple residue pairs. cases, the inverted indexing strategy” “recent PDB deposited depositions”, “6,814,159,549 residues pairs”

Reviewer #2: The manuscript describes a new technique for searching 3D motifs into structures.

The idea behind this method is nice and the calculation time very limited, especially because the search is performed using residue identity and because the tolerance in geometric differences is kept low.

The manuscript is interesting and well written, nevertheless I see some weak points that I will try to summarize below.

First of all, the only dataset that that manuscript is using to assess the performance of the method is the output of another method, Fit3D. Fit3D is an exhaustive method, but being exhaustive is not being perfect. Every method has its own parameters and performances, therefore I think this point is weak and the work should be strengthened by comparing the results with data extracted from biological databases (PROSITE, with its structural appendix, and other DBs).

The Authors report the number of false negatives that the searches give, never reporting also the false positives. I think that a fair evaluation of this work should contain also MCC or other standard more comprehensive parameters.

I feel very puzzled by the description that the Authors choose for all the residue pairs. They report the identity of the two residues (i.e. DS for aspartic acid and serine) and then 3 integer numbers associated to the backbone distance (Calpha for amino acids), the side-chain distance (Cbeta for amino acids) and an angle defined by the two vectors connecting backbone and side-chain of these two residues. I do not like the choice of Cbeta as representative of any side-chain (from Trp to glycine), and I am not sure that the descriptor of the residue pair is symmetric with respect to the same pair positioned in a different order in the sequence, even if in the same relative position in space. I think that if this is true, this method would be unable to identify identities of 3D motifs in non-homologous proteins. Or also 3D motifs with residues in different chains, situated in reverse order in the PDB file.

A minor weakness of this method is that it works with residue identities, while biologically meaningful 3D motifs usually also allow similar residues in the same positions.

Reviewer #3: In this paper authors describe a novel approach for structural motif detection and retrieval based on an inverted index strategy exploiting a simplified geometrical description of structural motifs. The advantages of this method are substantial since it enables fast and effective structural motif search over the entire PDB archive in a few seconds and with a very low FNR.

Results are presented covering five different queries discussing potentials and drawbacks of the method case-by-case. Overall, the manuscript is very well-written and it provides a sound contribution to the field.

I have only two minor comments:

1. In all presented test cases, authors have chosen a specific configuration for each query taken from specific PDB structures. I assume these configurations are the most common for each motif examined. However, I believe that analyzing how the choice the query configuration influences results would definitely add to the paper. The analysis could be also restricted to a single motif e.g. His-Asp-Ser motif.

2. In Tables 1 and S1 authors report search results for the five case studies. I suggest to also include results obtained with other approaches e.g. using the Fit3D method. This would provide the reader with a side-by-side comparison that would better highlight the advantages (in terms of time, hits and FNR) of the proposed approach over exhaustive search strategies.

**Have all data underlying the figures and results presented in the manuscript been provided?**

Reviewer #1: Yes

Reviewer #2: Yes

Reviewer #3: Yes

PLOS authors have the option to publish the peer review history of their article (what does this mean?). If published, this will include your full peer review and any attached files.

Reviewer #1: No

Reviewer #2: No

Reviewer #3: No
---

## [Decision Letter · Decision Letter 1]

9 Nov 2020

Dear Bittrich,

We are pleased to inform you that your manuscript 'Real-time structural motif searching in proteins using an inverted index strategy' has been provisionally accepted for publication in PLOS Computational Biology.

Best regards,

Marco Punta

Associate Editor

PLOS Computational Biology

Arne Elofsson

Deputy Editor

PLOS Computational Biology

Reviewer's Responses to Questions

**Comments to the Authors:**

Reviewer #2: I appreciated Authors answers (especially the one to Remark3), and I am afraid that I have to accept the answer to Remark1

Reviewer #3: Authors answered to all my concerns in the revised version of manuscript. I have no further comments to do.

**Have all data underlying the figures and results presented in the manuscript been provided?**

Reviewer #2: None

Reviewer #3: Yes

PLOS authors have the option to publish the peer review history of their article (what does this mean?). If published, this will include your full peer review and any attached files.

Reviewer #2: No

Reviewer #3: No

---

## [Editor Report · Acceptance letter]

1 Dec 2020

PCOMPBIOL-D-20-00730R1 

Real-time structural motif searching in proteins using an inverted index strategy

Dear Dr Bittrich,

I am pleased to inform you that your manuscript has been formally accepted for publication in PLOS Computational Biology. Your manuscript is now with our production department and you will be notified of the publication date in due course.

With kind regards,

Nicola Davies
